

# Caffeine vs Carbamazepine as indicators for wastewater pollution in a karst aquifer

Noam Zach Dvory[1,2], Yakov Livshitz[3], Michael Kuznetsov[1], Eilon Adar[1], Guy Gasser[4,5], Irena Pankratov[4], Ovadia Lev[5] and Alexander Yakirevich[1]

[1]Zuckerberg Institute for Water Research, J. Blaustein Institutes for Desert Research, Ben-Gurion University of the Negev, Israel
     [2]Etgar A. Engineering Ltd.
     [3]Israel Hydrological Service, Israel Water Authority, Israel
     [4]Water Monitoring Laboratory, Israel Water Authority, Israel
[5]The Hebrew University, Israel

*Correspondence to*: Noam Z. Dvory (nzd@etgar-eng.com)

**Abstract.** This paper presents the analysis of caffeine and carbamazepine transport in the subsurface as a result of wastewater release in the Sorek creek over the outcrops of the Yarkon-Taninim aquifer in Israel. Both caffeine and
carbamazepine were used as indicators for sewage contamination in the subsurface. While carbamazepine is considered conservative, caffeine is subject to sorption and degradation. The objective of the study was to quantify differences in their transport under similar conditions in the karst aquifer. Water flow and pollutant transport in a 'vadose zone - aquifer' system were simulated by a quasi-3D dual permeability numerical model. The results of this study show that each of these two pollutants can be considered as effective tracers for characterization and assessment of aquifer contamination.
Carbamazepine was found to be more suitable to assess the contamination boundaries, while caffeine can be used as a contaminant tracer only briefly after contamination occurs. In instances where there are low concentrations of carbamazepine which appear as background contamination in an aquifer, caffeine might serve as a better marker for detecting new contamination events, given its temporal nature. The estimated caffeine degradation rate and the distribution coefficient of linear sorption isotherm were 0.091 $d^{-1}$ and 0.1 L/kg, respectively, which imply a high attenuation capacity. The results of
the simulation indicate that by the end of the year most of the carbamazepine mass (approximately 95 %) remained in the matrix of the vadose zone, while all of the caffeine was completely degraded a few months after the sewage was discharged.

## 1. Introduction

Sewage infiltration into the subsurface can cause groundwater pollution. Carbonate aquifers present a higher risk for groundwater quality contamination due to the presence of preferential flow paths. Predicting and quantifying sewage
infiltration and transport in carbonate aquifers is complicated due to exchanges between slow flow in the matrix and fast flow in conduits (Geyer et al., 2007).





The micropollutants carbamazepine (CBZ) and caffeine (CAF) are both widely used as indicators for anthropogenic contamination in groundwater (Seiler et al., 1999). CBZ is generally accepted as a stable indicator for untreated/treated sewage (Clara et al., 2004; Dvory et al., 2018a; Fenz et al., 2005; Gasser et al., 2010). CAF concentration is often higher than CBZ concentration at the source (sewage) and consequently detected in soils, sediments (Bradley et al., 2007;

Klabunde, 2016), surface water (Bueno et al., 2010; Chen et al., 2006; Ferreira, 2005; Kolpin et al., 2004), lakes and seawater (Buerge et al., 2003; Gardinali and Zhao, 2002; Knee et al., 2010). Additionally, several studies have detected higher concentrations of CAF, as opposed to CBZ, in the groundwater as well (Godfrey et al., 2007; Lapworth et al., 2015; Manamsa et al., 2015; Metcalfe et al., 2011). CAF, like other micropollutants, is subject to degradation and sorption. These processes reflect major mechanisms for CAF attenuation in the environment (Hillebrand et al., 2015) and raise doubts as to

its efficacy as a tracer for detecting and quantifying contamination from wastewater (Ahmed et al., 2008; Liu et al., 2014). The above implies that CAF concentration is often higher and easier to detect than CBZ close to the pollution source briefly after contamination occurs. As a result, CAF was found to be a possible indicator for sewage in rapid flow systems, such as karst aquifers (Hillebrand et al., 2012b).

CAF attenuates in the subsurface by biodegradation and sorption processes (Martínez-Hernández et al., 2016). Those

processes are affected by local environmental conditions. Recent studies reveal fast biodegradation rates in carbonate aquifer conduits (Hillebrand et al., 2015, 2012b). In such aquifers, the flow is also affected by the connections between the matrix and conduit flow paths, which can influence the CAF attenuation. Lab (Arye et al., 2011; Conn and Siegrist, 2009; Hebig et al., 2017; Martínez-Hernández et al., 2016) and field (Hillebrand et al., 2012b; Zhang et al., 2013) experiments were carried out by many researchers in order to assess CAF sorption. Sorption parameters can vary as a result of the groundwater hosting

media. Hebig et al. (2017) showed that full removal of CAF was observed in the presence of organic carbon and no sorption was detected in iron-coated sand. The attenuation of CAF in the unsaturated zone can be very high, as shown by Martínez-Hernandez et al. (2017) by column test experiments and simulations.

Until the current study, simulations of micropollutant transport in the subsurface were mostly performed with either single (Bertelkamp et al., 2014) or dual porosity (Geyer et al., 2007, Hillebrand et al., 2012a,b) one-dimensional models for

saturated or unsaturated (Martínez-Hernandez et al., 2017) conditions. However, under field scale conditions, the fate of contaminants is affected by both the transport through vadose zone and lateral spread in the groundwater.

The objective of this study was to assess differences in CAF and CBZ attenuation by simulating their transport in the karst/fractured-porous unsaturated zone and groundwater system with a dual permeability mathematical model as described by Dvory et al. (2018a). The sorption and degradation parameters for CAF were estimated using observed concentrations in

a single well (EK11). Simulation results allowed for the characterization of CAF transportation and natural attenuation processes from the initial release of sewage at the Sorek creek bed, its infiltration into the unsaturated zone and transport into the aquifer. The study employed field observations and simulations of CAF and CBZ transport to reveal the effect of factors leading to differences in the attenuation of each micropollutant.





The full description of the field experiment, the mathematical model development, and the simulation results of CBZ transport are presented in Dvory et al. (2018a). In this paper we provide a short description of those processes, for the convenience of readers, and include only details which are essential for understanding of the presented material.

## 2. Materials and methods

### 2.1. Sewage release event

The Sorek creek watershed (approximately 88 km$^2$ in study area) is located west of the city of Jerusalem, Israel (Fig. 1). A local reservoir (Beit Zait), located 2.05 km upstream from the study site, collects surface flow and limits the natural runoff downstream. Periodically, the reservoir discharges its reserves; once every few rainy seasons and regular controlled releases from the dam downstream (Dvory et al., 2018b).

The geology of the area is comprised of a carbonate section of the Judea group (Dvory et al., 2016). The unsaturated zone is thick, spanning tens to two hundred meters. The groundwater primarily flows in the south-west direction (Dvory et al., 2016).

The current study examined a discharge event, which took place between April 2-19, 2013, when wastewater was released from a main sewage pipeline on five separate occasions into the Sorek creek (Fig 1; Fig. 2c) (Dvory et al., 2018a). CBZ

served as an indicator for the identification and quantification of sewage water migration into the aquifer (Dvory et al., 2018a) and CAF was used to assess its attenuation and suitability as a tracer for wastewater contamination characterization.

### 2.2. Field work

Field work, including water sampling and hydrological monitoring, was done to monitor the distribution of the discharged sewage (Dvory et al., 2018a). Over the course of 310 days, twenty three groundwater samples were taken from a depth of

100 m below the ground surface. The intervals between sampling events ranged from one to fifty six days, where the interval was shorter during the expected tracer breakthrough time in order to provide a higher temporal resolution of the CBZ and CAF tracer breakthrough curves (BTCs). Additionally, a data logger was installed in the observation well EK11 (Fig. 1) in order to take groundwater level and temperature data measurements every 30 minutes (Solinst Levelogger). Hourly measured values for precipitation and evaporation rates were acquired from a local the Israel Meteorological Service (IMS)

weather station ("Tzuba Station"). Data on sewage and surface runoff discharge rates were obtained from gauging stations (Fig. 1).



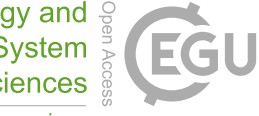

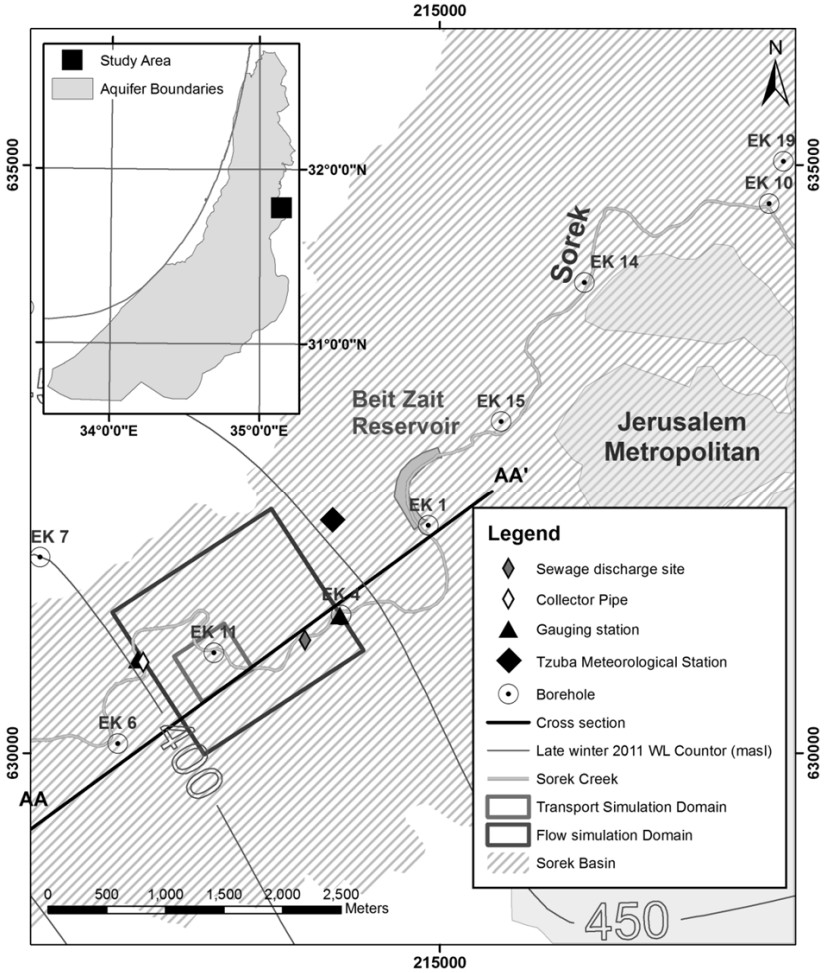

Figure 1: The upper Sorek Basin monitoring sites and flow and transport simulation domains (after Dvory et al., 2018a; aquifer boundaries from Dafny, 2009).





### 2.3. Analytical methods

A methodology using the Agilent G6410A Triple Quadrupole mass spectrometer (QQQ) with electrospray ionization ion source (ESI) was used in order to identify and verify CBZ and CAF presence in collected water samples. The LC/MS/MS method's main characteristics are presented in Table 1. The method was developed from the EPA Method 1694 guidelines

for acid and basic compounds elution from solutions containing less than 1% solids.

Measured amounts of labelled compounds (CBZ-d10 and CAF 13C3), were added to the water samples. The compounds were then extracted by Solid Phase Extraction (SPE) with Oasis HLB 60 mg cartridges (Waters, Milford, MA) using 1000 mL of each sample. Analytes were subsequently eluted with methanol and formic acid solutions, and the mixed extracts were concentrated to a final volume of 5 mL by nitrogen flow. Analytes were separated with an Agilent ZORBAX Eclipse Plus

C18 (2.1 mm ID, 100 mm length, 3.5 µm particle size). Column temperature was set at 25 $^{\circ}$ C. The mobile phase consisted of 10 % Acetonitrile, 90 % H$_2$O, and 0.1 % formic acid. The eluent composition included: initial conditions, 10 % Acetonitrile fed at 0.2 ml/min for 5 minutes. After 6 minutes, the flow rate was increased to 0.3 ml/min, and eluent composition was changed to 60 % Acetonitrile, fed at a flow rate of 0.3 ml/min until 24 minutes. In the last stage, eluent composition was gradually increased until it reached 100 % Acetonitrile at 30 minutes. Injection volume was 15 µl (Ferrer et

al., 2008).

Quantifications were carried out by isotope labelled internal standards for CBZ and CAF, by multipoint calibrations. Limits of quantification (LOQs) are shown in Table 1 and were calculated at 10 times the background levels, along with the recovery at 1000 ng/l. The linearity of the response of three orders of magnitude was demonstrated ($R^2$>0.99) for both the pharmaceuticals studied.

### 2.4. Modelling

### 2.4.1. Numerical model

Variable saturation flow and contaminant transport in the vadose zone and the groundwater were simulated using a quasi-3D model. Two overlapping continua representing highly conductive karst/fractures (conduits) (*c*) and the low permeability matrix (*m*) were used to simulate the karst/fractured-porous medium. The quasi-3D approach (Levy et al., 2017; Twarakavi

et al., 2008; Yakirevich et al., 1998) was used as the basis for the numerical model. The model uses a series of 1D equations in a variably-saturated zone to simulate the 'vadose zone – aquifer' system and 3D equations for saturated flow and transport to simulate groundwater. At the dynamic phreatic surface the 1D and 3D equations are coupled (Kuznetsov et al., 2012). Unsaturated flow in high and low permeability regions is described by two Richards' equations accounting for the linear exchange kinetic between them (Dvory et al., 2016). Horizontal flow in the vadose zone was neglected. The pollutant

transport equations in conduits and matrix respectively are:





$$r_c \left[ \frac{\partial(\theta_c C_c)}{\partial t} + \rho_b \frac{\partial F_c}{\partial t} \right] = \nabla \cdot \left[ r_c \left( \mathbf{D}_c \theta_c \nabla C_c - \mathbf{q}_c C_c \right) \right] - M_{cm} - \lambda r_c \left( \theta_c C_c + \rho_b F_c \right) \tag{1}$$

$$(1 - r_c) \left[ \frac{\partial(\theta_m C_m)}{\partial t} + \rho_b \frac{\partial F_m}{\partial t} \right] = \frac{\partial}{\partial z} \left[ (1 - r_c) \left( D_m \theta_m \frac{\partial C_m}{\partial z} - q_{zm} C_m \right) \right] + M_{cm} - \lambda (1 - r_f)(\theta_m C_m + \rho_b F_m) \tag{2}$$

where $C_i$ and $F_i$ are the concentrations in liquid and sorbed phases, respectively, $i=c$ and $m$ for conduit and porous matrix,

$\theta_i$ is water content; $\rho_b$ is the rock bulk density; $\mathbf{D}_i$ is the hydrodynamic dispersion tensor; $\mathbf{q}_i$ is the Darcy's flux of water;

$q_{zm}$ is the vertical water flux in blocks, $M_{cm}$ is a term which accounts for the solute exchange between conduits and porous

matrix, respectively; $\lambda$ is the degradation rate; $r_c$ is the relative conduits volume; $z$ is the vertical coordinate, and $t$ is time.

Under field scale conditions, when larger scale hydrological models are applied, there is not usually a detailed quantitative

knowledge of these controlling factors and simple 1st-order degradation is often assumed (Bradlay et al, 2007). Therefore, in

the present investigation the degradation rate coefficient for conduits and matrix is assumed to be the same.

Sorption of a solute is described by the following linear isotherm:

$$F_i = K_D C_i \tag{3}$$

where $K_D$ is the distribution coefficient.

At time zero, the initial flow condition prescribes the pressure head or water content distribution along the simulation profile.

Temporal flow fluxes or heads are defined by boundary conditions. The changes in concentration are defined at the inflow

boundaries; while zero concentration gradient was prescribed at the outflow boundaries. Initial concentrations for CBZ and

CAF were assigned throughout the entire simulation domain. The distribution of CBZ initial concentrations was discussed in

Dvory et al. (2018a), and based on several measurements. A zero level concentration for CAF was used for the entire

domain.

### 2.4.2. Mathematical model setup

The aforementioned equations were solved using a method of finite differences. The MODFLOW model (Harbaugh et al.,

2000) was modified to incorporate 1D Richards' equations (Kuznetsov et al., 2012) and to account for a double permeability

approach. In order to simulate solute transport in the vadose zone and groundwater, the MT3D (Zheng and Wang, 1999)

numerical code was also modified. Pre- and post-processing data was conducted with the Groundwater Modelling Software

(GMS 6.0, 2002).

A three step approach was used to address flow and transport problems: 1) flow was simulated in a large domain with a

coarse grid and well-defined hydrogeological boundaries; 2) flow and CBZ transport were simulated in a small domain,

using a more refined grid (Fig. 1); and 3) flow and CAF transport were simulated in the small domain. This sequential

process was used in order to minimize processing time and increase the accuracy of the solution, owing to a lack of central



processing unit (CPU) memory. Both the large and small domains had a uniform grid on the horizontal plane and a variable size grid in the vertical plane, which varied in respect to the ground level and aquifer base altitudes. The large domain was 1600×1400 m on the horizontal plane and varied from 170 m to 445 m on the vertical plane. The grid size was 20×20 m on the horizontal plane. The vertical plane of the grid was composed of 38 layers which increased from 0.0002 L to 0.1 L

(L(x,y) is the aquifer thickness including the unsaturated zone). Simulations in the large domain were used to calibrate hydraulic parameters and the western lateral boundary condition for the flow model.

The size of the small domain was 590×460 m on the horizontal plane and varied from 185 m to 280 m on the vertical plane. The grid was uniform 10×10 m on the horizontal plane and the vertical plain was the same as for the grid in the larger domain. The transient boundary conditions for the small domain were obtained from the solution from the larger domain

(GMS 6.0 Tutorials, vol. 2).

### 2.4.3. Calibration and sensitivity analysis

The model flow component was calibrated to fit the observed aquifer water levels in EK11. As a result, a set of hydraulic parameters was estimated (Dvory et. al., 2016, 2018a). The PEST software (Doherty, 2004) was used to calibrate the transport component of the model by minimizing the least squared errors between simulated and observed concentrations of

CBZ and CAF in EK11. First, the CBZ breakthrough curve was used to define longitudinal dispersivity ($a_L$), ratios of transverse to longitudinal dispersivities and ratios of vertical to longitudinal dispersivities ($a_T/a_L$ and $a_Z/a_L$), exchange rate parameter ($\eta_c$), and the distribution coefficient ($K_D$) for CBZ sorption, assuming zero degradation rate of CBZ. Then, the distribution coefficient for CAF sorption ($K_D$) and its degradation rate ($\lambda$) were found using CAF concentration measurements.

An additional estimate of the CAF degradation rate in the karst aquifer was performed analytically using both normalized CBZ and CAF concentrations. By neglecting the differences in sorption of both contaminants, the CAF degradation rate was found by minimizing following expression:

$$\min_{\lambda} \sum_{j=1}^{n} \left[ \ln\left( \frac{\overline{C}_{CAF,j}}{\overline{C}_{CBZ,j}} \right) + \lambda t_j \right]^2 \tag{4}$$

where $\overline{C}_{CAF,j}$ and $\overline{C}_{CBZ,j}$ are relative to the observed concentrations of CAF and CBZ in the sewage at time $t_j$,

respectively. Expression (4) reflects the relative decrease of degradable CAF concentration compared to that of non-degradable CBZ.

The sensitivity analysis was performed in order to evaluate the impact of solute transport parameters on simulated concentrations. The analysis involved varying parameters one at a time from their best fit value in the calibrated model and comparing the root mean square errors (RMSE) of the obtained simulation results to the calibrated model's (Geyer et al.,

30  2007).



## 3. Results and discussion

### 3.1. Aquifer water level and pollutants concentration fluctuation

Observations revealed that the wastewater and surface water infiltration had an immediate impact on the water level in the aquifer and on contaminant concentrations (Fig. 2). Groundwater level increased as a result of winter precipitation, runoff

and four significant sewage release events. Groundwater level rose to 9.3 m during leakage events and water level decline was observed between events. Reservoir water infiltration triggered a further increase in the water level, reaching 16.6 m above the initial natural groundwater level. In the dry season there was a measured decrease in the groundwater level and the following winter, the aquifer water level rose again as a result of precipitation and a runoff infiltration caused by Beit Zait dam overflow.

CBZ concentration variation mirrored the aquifer water level fluctuation. On April 7, 2013, CBZ concentration in sewage was measured at 995 ng/l. During the sewage release events, CBZ concentrations in the aquifer rose to 21.0-25.9 ng/l from background levels of 3.7-7.0 ng/l in EK11 (Fig 2). Later, due to releases from the Beit Zait dam into the creek, CBZ concentration rose to 47.0 ng/l. During the dry season CBZ concentration returned to background levels and rose again, to 11.0 ng/l, by the end of year due to runoff infiltration.

CAF was not initially detected in EK11 (July 4, 2012) but was detected later on, likely because of the subsequent sewage spillages on February 15, 2013 and April 2, 2013. Prior to the dam opening on February 15, 2013 and April 23, 2013 CAF concentrations in the reservoir were 100 ng/l and 245 ng/l, respectively. In the same period (April 7, 2013), CAF concentration in sewage was 58,500 ng/l. Three main CAF concentration peaks were observed in groundwater samples during the sewage discharge event (100.0 ng/l, 300.0 ng/l and 240.0 ng/l respectively). The first peak was related to sewage

discharge and the last two CAF peaks were connected to the controlled releases from the Beit Zait dam. CAF concentration level declined to 40.0 ng/l between the sewage leakage episodes and the reservoir water discharge event. In addition, the last two CAF concentration peaks were higher than the first, both in scale and duration. This indicates that the reservoir water effectively flushed pollution from the thick vadose zone. In the dry season, CAF concentration declined to 0.01 ng/l as detected on May 26, 2013 and was not detected afterward.





**Figure 2: Time series data observation and calculation.**

**(A) Tzuba Meteorological station daily precipitation rate; (B) Dam runoff flow; (C) Sewage surface flow;**

**(D) Measured and simulated aquifer water level at EK11. (E) Measured CBZ and CAF concentration.**

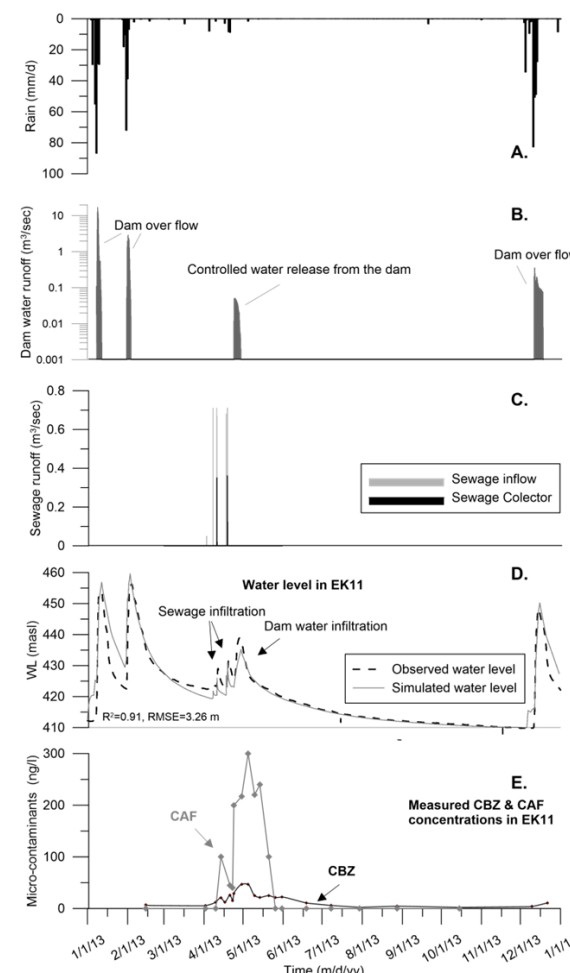



## 3.2. Modelling outcome

Simulated and measured concentrations of CBZ and CAF in EK11, actual and relative, are shown in Fig. 3a and 3b, respectively. Fair agreement between observed and simulated concentration was obtained. The model correctly predicted the major peaks of both CBZ and CAF concentration, with a root mean square errors (RMSE) of 7.4 ng/l and 53.9 ng/l,

respectively. The larger RMSE of CAF (compared to that of CBZ) is due to higher CAF concentration oscillation. The





model was not able to mimic a few significant fluctuations in CAF concentration. Despite the fact that CAF concentration (58,500 ng/l) in discharged sewage was around 59 times higher than CBZ (995 ng/l), the observed maximum concentration of CAF was only 6.4 times greater (Fig. 3a). Thus, there is a more significant reduction of CAF's relative concentration (Fig. 3b). Assuming no significant differences between conduit and matrix solute exchange, a drastic reduction in relative CAF

concentration can be attributed to sorption and degradation.

CBZ and CAF calibrated transport model parameters are presented in Table 2. A longitudinal dispersivity value of $a_L$=6.44 m lines with its expected field scale order of magnitude (Neuman, 1990). CBZ and CAF distribution coefficients ($K_D$) values were 0.011 l/kg and 0.1 l/kg respectively. This indicates a higher sorption of CAF than CBZ. In laboratory batch studies with riverbed sediments, Lin et al. (2010) showed that the sorption of CAF better fit the Freundlich isotherm, and its sorption was

highest amongst other pharmaceutical compounds studied. A batch test by Martínez-Hernández et al. (2016) revealed that sorption played a key role during the first 48 hours of contact with the soil, and gave way to biodegradation afterwards, with the fastest initial sorption velocities of CAF. Nevertheless, our results indicate that using linear sorption in simulating transport of micropollutants on a large field scale can also be appropriate.

The degradation rate of CAF, found by inverse solution, was 0.091 d$^{-1}$. The value of this parameter, which was estimated

analytically using the expression (4), was 0.082 d$^{-1}$ which is a bit smaller than the result found by inverse solution. However, the analytical procedure does not account for the effect of CAF sorption on concentration. Porous aquifer biodegradation rate for CAF was evaluated by Swartz et al. (2006) as 0.07-0.014 d$^{-1}$. Bradley et al. (2007) showed that the rate of CAF biotransformation in stream water is sensitive to in situ redox conditions (0.72-3.14 d$^{-1}$) and Martínez-Hernández et al. (2016) calculated a similar value range in laboratory batch experiments (0.37-4.18 d$^{-1}$). The role of different redox conditions

and biodegradable dissolved organic carbon on CAF biodegradation was examined in soil column experiments by Regnery et al. (2015). They estimated that CAF biodegradation rate was 0.37 d$^{-1}$ for oxic redox conditions when biodegradable dissolved organic carbon was present. Hillebrand et al. (2015, 2012b) evaluated CAF biodegradation rates between 0.16 and 0.19 d$^{-1}$ based on tracer experiments in karst conduits, which is similar within an order of magnitude from the value obtained in this study. The above studies showed that CAF degradation rate is expected to be higher in oxic carbonate aquifer rapid

flow systems.





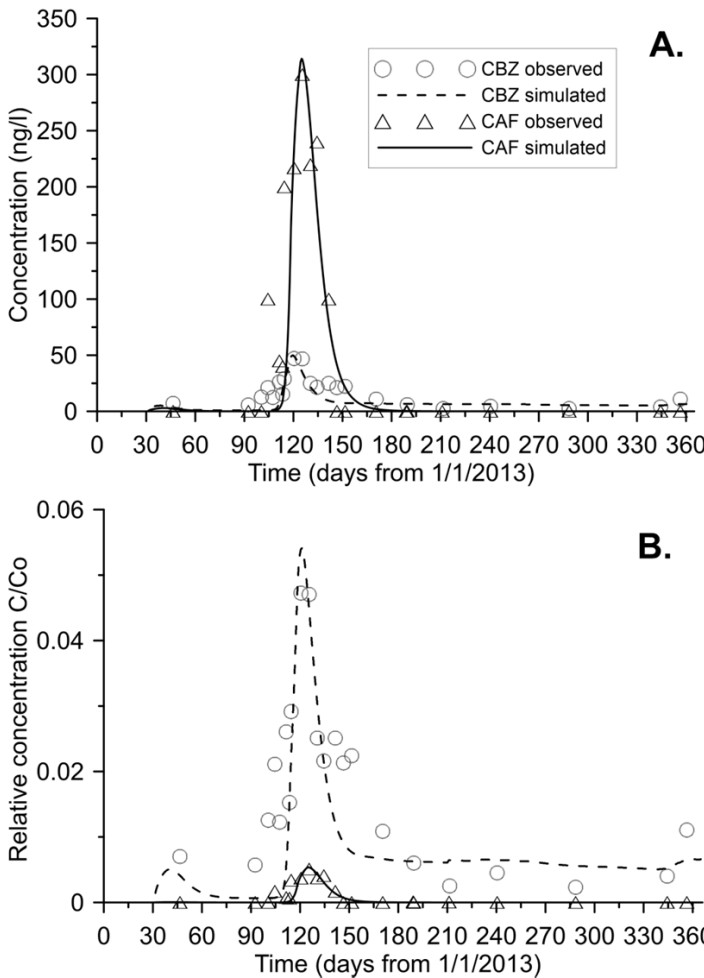

**Figure 3:** **(A) Observed and simulated BTCs of CBZ and CAF in EK11; (B) Relative concentration variations of CBZ and CAF in EK11.**



### 3.3. Sensitivity analysis

The sensitivity of the modelled CBZ concentrations with respect to the parameters of dispersivity and the solute exchange rate between conduits and matrix was discussed in Dvory et al. (2018a). This analysis is also valid for CAF. The sensitivity of the model to the CAF degradation rate ($\lambda$) and the distribution coefficient ($K_D$) was assessed by changing each parameter

by 10 % and 25 % of their calibrated values (Fig. 4). The model was found to be sensitive to variations in both parameters. A decrease in $\lambda$ by 10 % and 25 % decreases peak CAF concentration by around 18 % and 54 %, respectively. Similar increases in $\lambda$ decreases the maximum concentration by 15 % and 34 %, respectively. Increasing or decreasing the $\lambda$ fitted value results in a narrower or wider breakthrough curve, respectively (Fig. 4a).

One of the model assumptions was to prescribe the same pollutant degradation rate in both conduits and matrix. In order to

evaluate the effect of CAF degradation in the matrix on the breakthrough curve, we performed an additional simulation by assigning $\lambda$ =0 in matrix, while keeping its calibrated value in conduits. The result shows that the simulated breakthrough is similar to the calibrated one, except the former has a non-zero concentration (around of 9 ng/l) tail which is caused by solute exchange between the matrix and conduits. Observations show that only a few months after sewage was discharged into the creek CAF concentration was no longer detected in EK11, therefore, we hypothesize that degradation of CAF occurs in both

the matrix and conduits. However, based on the existing dataset, we cannot evaluate the differences between them.

A similar sensitivity trend is observed with respect to the distribution coefficient $K_D$. A decrease in $K_D$ by 10 % and 25 % leads to consequent decrease in peak concentration by 16 % and 43 %, respectively. An increase in $K_D$ by 10 % and 25 % leads to a 14 % and 31 % increase, respectively. This also causes a delay in peak concentration arrival time by around 1 and 3 days, respectively, compared with the calibrated case.






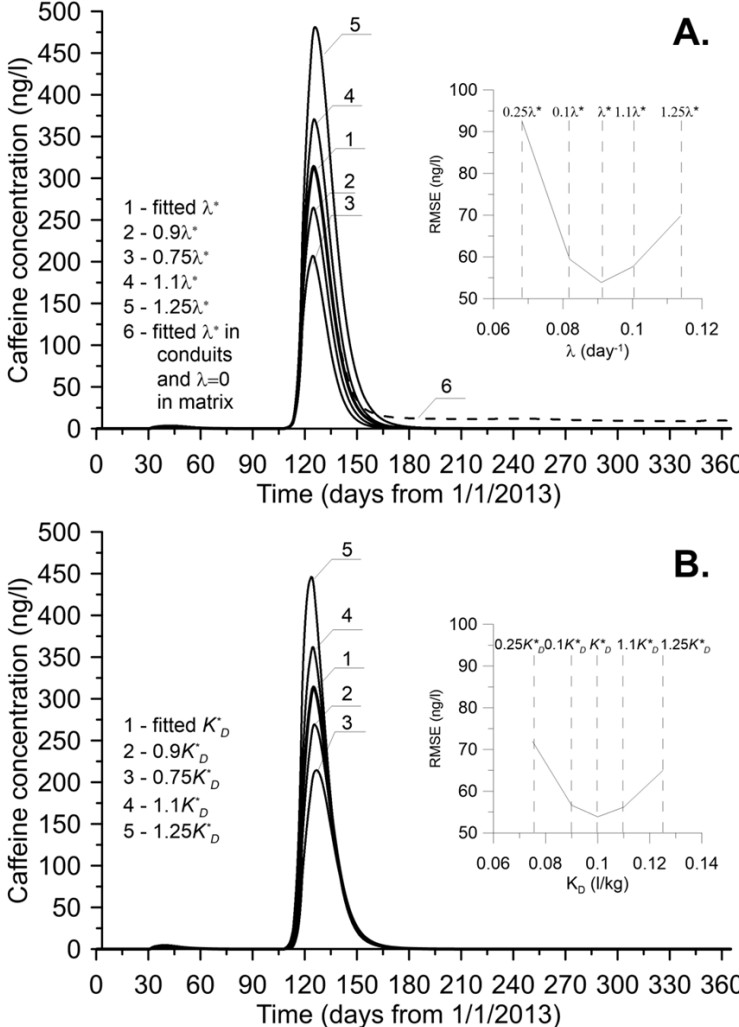

**Figure 4: Simulated CAF sensitivity to parameters changes (A) the degradation rate and (B) the distribution coefficient.**





### 3.4. CAF vs CBZ as indicators for sewage contamination in a carbonate aquifer

Both CBZ and CAF can serve as tracers to monitor sewage contamination in carbonate aquifers. Each of these contaminants has specific advantages as a pollution indicator. CBZ is more stable than CAF, and this characteristic enables CBZ to migrate far from the pollution source into the aquifer. Thus, CBZ can be used to estimate the reach of sewage pollution in the

aquifer. However, there are often low background levels of CBZ, which makes it hard to distinguish a new pollution event (Hillebrand et al., 2012). The advantage of CAF as an indicator is its higher concentration levels proximate (in time and location) to the event. The simulation results presented in Fig 3b elucidate the difference between CBZ and CAF transport and attenuation in the aquifer. Several weeks after sewage was discharged CAF was no longer detected in groundwater near the contaminant source. Dissimilarly from the CAF breakthrough curve, which shows a rapid decline in concentration, the

CBZ breakthrough curve exhibits a long tail of low concentration which can increase during the rainy season due to enhanced exchange between porous matrix and conduits. The simulation results demonstrated that by the end of year, a relatively high volume of CBZ (around 95% of discharged) remained in the vadose zone matrix close to the source , and tens of meters downstream in the groundwater; while all the CAF was degraded soon after sewage discharge stopped.

In the present study, the half-life of CAF was calculated to be 7.6 days. This result was similar, within an order of

magnitude, however almost twice as long as the half-life estimated by Hillebrand et al. (2015, 2012b) which ranged between 3.7 to 4.3 days. However, the present study accounts for simulation of flow and transport in a thick unsaturated zone in both the conduits and matrix. Other conditions affecting CAF attenuation were also different.

The sources of uncertainty in the mathematical model predictions were discussed by Dvory et al. (2018a). For example, due to technical problems, CBZ and CAF concentrations were measured only in one monitoring well in the current study. Better

model calibration could be obtained if data was available from other monitoring wells and used as inputs into the transport model.

### 4. Conclusion

A Quasi 3D dual permeability mathematical model was used to simulate CAF transport and attenuation in the vadose and in the saturated zone of the studied carbonate aquifer. The CAF sorption and degradation rate parameters were found by

calibrating with monitoring data from a single well. Simulation results from the model were compared to previous simulations of CBZ transport on the site (Dvory et al., 2018a). Both pollutants were found as effective tracers for characterization and assessment of aquifer contamination by wastewater. CBZ was found to be more suitable to assess the contamination boundaries; while CAF can be used as a contaminant tracer only shortly after a contamination event occurs. In addition, CAF's relatively ephemeral nature could present an advantage when trying to detect new contamination events in

aquifers that have background CBZ contamination.

The estimated degradation rate and the distribution coefficient of CAF were 0.091 d$^{-1}$ and 0.1 L/kg, respectively, which may explain its high attenuation potential in the studied aquifer. The calculated CAF degradation rate was slightly smaller



compared with previous studies of CAF decay in karst aquifers. The sensitivity analysis results revealed that the model is highly sensitive to the CAF degradation rate and the distribution coefficient of linear sorption.

The simulations reveal that by the end of year, about 95% of CBZ mass remained in the vadose zone porous matrix near the pollution source and tens of meters downstream in the groundwater; while all the CAF was degraded a few months after
leakage stopped.

**Author Contributions:** Noam Zach Dvory, Yakov Livshitz, Eilon Adar and Alexander Yakirevich conceived of and designed the research; Alexander Yakirevich supervised the study; Guy Gasser, Irena Pankratov and Ovadia Lev supervised and preformed the lab analysis; Noam Zach Dvory collected and analysed the data; Michael Kuznetsov developed the
numerical code; Noam Zach Dvory wrote and edited the paper; Alexander Yakirevich, Yakov Livshitz and Michael Kuznetsov reviewed the manuscript.

**Conflicts of Interest:** The authors declare no conflict of interest.

**Acknowledgements**

The research was funded by the Israeli Water Authority, grant number 4500088042. The authors thank Mr. Guy Reshef and Dr. Gabi Weinberger for fruitful discussions and help.

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




**Tables**

| | Precursor ion | Product ions, quantifier and (qualifier) | Fragmentation voltage (V) | Collision energy (eV) | Electrospray ionization (ESI) | Mean recovery,% (±standard deviation) | LOQ (ng/l) |
|---|---|---|---|---|---|---|---|
| **CBZ** | 237 | 194 (192) (179) | 120 | 15 10 35 | positive | 86.2 ± 14.2 | 1 |
| **CAF** | 194.9 | 137.7 (109.8) | 110 | 20 25 | positive | 82.1 ± 10.2 | 2 |
| **CBZ-d$_{10}$** | 247 | 204 (202) | 120 | 15 10 | positive | | - |
| **CAF $^{13}$C$_3$** | 197.9 | 139.7 (111.8) | 110 | 20 25 | positive | | - |

**Table 1: LC/MS/MS method main characteristics for CBZ and CAF**

| Parameter | CBZ | CAF |
|---|---|---|
| | | 10 |
| Molecular diffusion coefficient in water[1] (m$^2$/d) | 4.8·10$^{-5}$ | 4.2·10$^{-5}$ |
| Karst/fracture (conduit) longitudinal dispersivity[2] (m) | 6.44 | |
| Transverse/longitudinal dispersivities ratio[2] | 0.001 | |
| Vertical/longitudinal dispersivities ratio[2] | 0.1 | 15 |
| Solute exchange parameter between conduits and matrix[2], (1/m$^2$) | 0.078 | |
| Distribution coefficient, $K_D$ (l/kg) | 0.011 | 0.1 |
| Rate of degradation (d$^{-1}$) | 0.0 | 0.091 |

20

[1]Calculated using molecular weight of CBZ (Schwarzenbach et al., 1993).
[2]Dvory et al (2018a).
**Table 2: Transport model parameters**