# Peer review of "Caffeine vs Carbamazepine as indicators for wastewater pollution in a karst aquifer"

_Hydrology and Earth System Sciences, 2018_

## Referee Comment (RC1) · Anonymous Referee #1 · 6 Sep 2018

Summary

The manuscript presents an elegant modeling effort of a sewage spill event in a wash overlaying a dual-domain (rock-blokes and conduits) karstic aquifer. The pharmaceutical Carbamazepine (CBZ) and the mostly beverage sourced (Coffee, Cola etc,) Caffeine (CAF) were used as the sewage tracers. CBZ simulated as a conservative tracer vs. CAF which adsorbs and follows first-order degradation. The pretty unique modeling system of the porous medium that is used in this paper, in which the vadose zone is modeled by 1D (vertical) dual-permeability sub-models, that are linked to the 3D sub-model of the aquifer was described in a previous paper were the conservative transport

of CBZ was analyzed. The addition of the calculation of the partition (distribution) coefficient and the first order degradation rate of CAF through calibration of this unique model and a relatively rare field set of data is the main and significant contribution of the paper. The results are Kd for CAF in dual permeability carbonate setting was 0.1 L/kg and the degradation rate 0.09 d-1. Practical conclusions were that CZB is a good tracer if you can analyze its small concentrations and have relatively sparse data in space and time around the sewage spill, whereas when data around the spill (space and time) exists the more common, zero background, and higher concentration CAF can serve as a good tracer of sewage spill as well.

Recommendation The paper is in some sense a follow up of Dvori et al., 2018. Nevertheless, the case study is so unique (sewage spill above thick karst system, level of surface and deep subsurface water sampling and monitoring, level of chemical analysis) and the modelling is advanced yet strait forward. And, above all, the added analysis in this work is an interesting and significant contribution to large field-scale reactive transport analysis. The paper is written good and to the point and will be of interest to varying level of karst hydrologist (as well as other flow and transport modelers, soil physicists etc,). Therefore, I recommend publication following minor (some very essential) revisions due to the following specific comments herein.

Specific comments

1) I will start with most annoying discrepancy and then write the comment chronologically as they appear in the manuscript. Perhaps I am wrong, but the authors should check very carefully if typo mistakes in the legend of Figure 4 messed the sensitivity analysis of Kd and Lamda in section 3.3. Yo the best of my understanding a breakthrough curve (BTC) of a degrading contaminant down gradient of an instantaneous spill should show a higher peek and a larger width for smaller degradation rates not for higher ones as shown Fig 4a shows. Check if BTC 5 and 3 were switched as well as BTC 2 and 4. The same for distribution coef. and Figure 4b: a BTC of a degrading and adsorbing contaminant will be shorter and retarded for a larger distribution coefficient
rather than a smaller one like it is in the figure (e.g. BTC 3). Check.

2) The graphics of figure 4 must be improved by showing a smaller time span so the area below the BTCs will be larger and retardation (fig 4b) and different tales (Fig 4a) will be visualized better.

3) P.1, L.14 - add carbonate before Yarkon-Taninim 4) P. 2 L. 30 – add Fig. 1 after EK11 (or delete EK11) 5) P. 3 L 22 – replace "data logger" with: pressure and temperature probe with data logging capability 6) P. 3 L25 add upstream and downstream from the well head after "stations" 7) Figure 1 the aquifer boundary inset – make it clearer for the fast reader. Add Tel Aviv location and or Mediterranean Sea, a north arrow etc., don't just send the international readership to lookup where is 35oE and 32oN. 8) P. 5 L. 19 change "pharmaceuticals" to micro pollutants or organic compounds or similar, caffeine is not a pharmaceutical. 9) P. 5 L. 29 – It would be appropriate to mention also Gerke and van Genuchten 1993 for the formulation of the dual permeability model. 10) P. 6 L. 4 – for consistency define qc (like you do for qzm) rather than qi 11)P. 6 L - I think the sentence in the beginning of the row would be better said as: Boundary conditions are of the type of transient head or transient flux. 12) P. 6 L 15 – Delete the sentence starting "Initial . . ." Its redundant. 13) P. 7 L. 13 should be parameters were rather than "was". 14)P 9 L. 29 – Delete "a" 15) P. 10 L. 2 – Delete "around" 16) P. 10 L. 10 change "amongst other" to" in comparison to 17) P. 10 L. 17 or 0.07 – 0.14 or 0.014-0.07 but not as written 18) P. 14 L. 13 "downstream" or downgradient 19) P. 14L. 15 "(2015, 2012b)" there is only 1 reference of Hillebrand et al. in the reference list 20) P. 14 L.18-21. Consider discarding, out of context and does contribute much.

Please also note the supplement to this comment:
https://www.hydrol-earth-syst-sci-discuss.net/hess-2018-426/hess-2018-426-RC1-supplement.pdf

426, 2018.

---

## Referee Comment (RC2) · Anonymous Referee #2 · 10 Sep 2018

**In its current form I would recommend publication with minor corrections**.

The authors studied transport dynamics of two tracers, carbamazepine CBZ and caffeine CAF, using a dual-continuum saturated/unsaturated flow and transport model including a coupled 1D model to simulate the vadose zone. The topic can be considered highly challenging, both from a conceptual point of view and the difficulties associated with the highly contrasting continua. The authors demonstrate the limitation and benefits of each substance. While CBZ due to its conservative nature can be used to delineate the extent of contamination it is affected by background concentrations. CAF on the other hand is affected by degradation but can be employed to identify short-term

contamination events. The paper is short but concise. In addition to the results from the Water Research paper the authors have addressed the transport of caffeine and compare it to the transport of carbamazepine and associated processes. Some of my comments target the specifics of the signal transformation, and if the authors address them, I believe the paper will stand out more compared to the WR paper.

I agree with the comments of the editor that the paper in parts resembles the paper published in Water Resources. Both manuscripts use the same (in most aspects) underlying flow and transport model and in that sense it is understandable that the authors use the model for various application cases (here most of the model description is omitted and the respective paper is cited for reference).

Figures 1,2 and 3b are taken from the previous manuscript with no or very little modification. In addition to the overlap as indicated by the editor this should be fixed (Figure 1 is probably the most critical one and should be slightly modified, also to avoid any copyright issues).

**Content-related comments:**

1. (p6): For the sake of completeness please add units to the description of the equations.

2. (p5): The section "Numerical model" would greatly benefit from a conceptual sketch of the model framework (the Water Research paper provides a conceptual model of the hydrogeological system only). The model is quite complex and as the authors have already limited the model description here. A conceptual sketch would also allow readers to understand more aspects of the model without first having to read another paper. This would also help to make the paper stand out slightly more compared to the WR paper.

3. (p5, line 28): From the description here it is not entirely clear to me if both the 1D and 3D part of the domain are subject to a multi-continuum coupling. In this sense also the terms high and low permeable region are (from a conceptual point of view) associated with different compartments of the aquifer. High permeable regions in the vadose zone are possibly (enlarged) fractures and to a limited degree former conduit systems depending on the longterm evolution of the system, while in the 3D part the high permeable regions are commonly the conduits. This should be clarified, possibly also in conjunction with my previous comment to add a conceptual sketch.

4. (p6): In addition to the conceptual sketch I think a figure showing the discretized model domain including boundary conditions (both for the large and small model) would be adequate to be added to the section "Mathematical model setup".

5. (p7, line 15-19): To what extent does the vadose zone possibly affect the (bulk) dispersivity? The chosen approach is common for saturated systems but may be affected by the vadose zone which imposes an additional transformation/dispersion of the signal. I understand that this is a very difficult topic and would only ask for a brief comment if this might be the case (or not if the authors can clearly rule this out). In this context the authors mention that CBZ is stored in the vadose zone (on page 14, line 11), hence I would expect an influence.

6. (p12): Are the parameters $\lambda$ and $K_d$ defined for both the 3D section and the 1D vadose zone? This information should be added. Please also see my comment 3.

   Is the degradation of caffeine affected by the presence of an airphase? If this is the case then this should be briefly discussed either here or in the discussion section.

7. (p13, Fig 4): I am a bit confused by both sensitivity analyses but may have missed some information in the manuscript. I would expect an opposite behavior for

$\lambda$ as well as $K_d$. I would expect lower peaks (and low tailing, i.e. generally a decline in mass) for higher values of degradation. The same applies for the distribution coefficient (which to my knowledge is commonly defined as activity of solid/aqueous phase). Here I would expect lower peaks for higher values of Kd as CAF tends to be in an sorped state.

In Figure 4a it is difficult to see where the peak of parameter combination 6 is (only the tailing is clearly visible). A different color (gray or colored) for the fitted values (both in A/B) could help to enhance visual clarity.

8. (p14, line 5): Is this correct? I would expect low background concentrations to be beneficial for the detection of a new signal.

**Typographic corrections:**

I am not a native speaker and can only partially comment on proper grammar. The following are mostly typographic corrections and recommendations to enhance the comprehensibility.

1. (p3, line 6): Maybe rephrase. Do the authors mean that Sorek creek watersheed accounts for 88km$^2$ within the study area or that the Sorek creek study area is 88km$^2$ in size?

2. (p3, line 19, 20): This may be journal-specific but commonly only numbers exceeding 12 are spelled out.

3. (p6-7, line 28/1): I assume the authors mean main memory not the CPU cache. Possibly rephrase as "owing to a lack of main memory".

4. (p7, line 4): Is the grid becoming finer towards the top or the bottom of the domain? Possibly rephrase to clarify.

5. (p12, line 11): "...assigning $\lambda = 0$ in the matrix,..."

6. (p13, Fig. 4): Please match the font/fontsize of the insets in A and B (lambda and Kd values)

7. (p14, line 23): "A quasi 3D dual permeability..."

8. (p14, line 25): I think it should be "calibration with monitoring data..."

---

## Short Comment (SC1) · 17 Sep 2018

Dear Referee #1, We would like to thank you for taking the time to carefully review our paper. We found your recommendations helpful and we think they will help enhance the paper and the accompanying graphs. We will address your comments and re-submit the paper soon. Thank you again, Noam

---

## Short Comment (SC2) · 17 Sep 2018

Thank you for your thoughtful and thorough review of our paper. We appreciate the time and effort you put into your comments. Your comments and questions were insightful and helped us determine where the paper needed further clarification and how to add to this article in order to differentiate it from previous publications. We will address your comments and re-submit the paper soon. Thank you again, Noam

---

## Author Comment (AC1) · 1 Oct 2018

Answers to Reviewer #1 comments:

General:

Thank you for the feedback. Your recommendations were helpful and insightful. All of the comments have been addressed, and the paper was edited accordingly.

Specific comments:

1) I will start with most annoying discrepancy and then write the comment chronologically as they appear in the manuscript. Perhaps I am wrong, but the authors should

check very carefully if typo mistakes in the legend of Figure 4 messed the sensitivity analysis of Kd and Lamda in section 3.3. To the best of my understanding a break-through curve (BTC) of a degrading contaminant down gradient of an instantaneous spill should show a higher peek and a larger width for smaller degradation rates not for higher ones as shown Fig 4a shows. Check if BTC 5 and 3 were switched as well as BTC 2 and 4. The same for distribution coef. and Figure 4b: a BTC of a degrading and adsorbing contaminant will be shorter and retarded for a larger distribution coefficient rather than a smaller one like it is in the figure (e.g. BTC 3). Check.

Answer: Thank you for this important comment. The legend in this figure was wrong indeed. We corrected both the figure and the relevant text.

2) The graphics of figure 4 must be improved by showing a smaller time span so the area below the BTCs will be larger and retardation (fig 4b) and different tales (Fig 4a) will be visualized better.

Answer: As suggested by the reviewer, the graphics have been revised.

3) P.1, L.14-add carbonate before Yarkon-Taninim

Answer: The correction has been made.

4) P.2L.30–add Fig. 1 after EK11 (or delete EK11)

Answer: The correction has been made.

5) P. 3 L 22 – replace "data logger" with: pressure and temperature probe with data logging capability

Answer: The correction has been made.

6) P. 3 L25 add upstream and downstream from the well head after "stations"

Answer: The correction has been made.

7) Figure 1 the aquifer boundary inset – make it clearer for the fast reader. Add Tel Aviv

location and or Mediterranean Sea, a north arrow etc., don't just send the international readership to lookup where is 35oE and 32oN.

Answer: As suggested by the reviewer, the graphics have been revised.

8) P. 5 L. 19 change "pharmaceuticals" to micro pollutants or organic compounds or similar, caffeine is not a pharmaceutical.

Answer: The correction has been made.

9) P. 5 L. 29 – It would be appropriate to mention also Gerke and van Genuchten 1993 for the formulation of the dual permeability model.

Answer: The correction has been made.

10) P. 6 L. 4 – for consistency define qc (like you do for qzm) rather than qi

Answer: The correction has been made.

11) P. 6 L - I think the sentence in the beginning of the row would be better said as: Boundary conditions are of the type of transient head or transient flux.

Answer: The correction has been made.

12) P. 6 L 15 – Delete the sentence starting "Initial ..." Its redundant.

Answer: The correction has been made.

13) P. 7 L. 13 should be parameters were rather than "was".

Answer: The word "was" refers to a (single) set of parameters. Therefore the suggested change was not made.

14)P 9 L. 29 – Delete "a"

Answer: The correction has been made.

15) P. 10 L. 2 – Delete "around"

Answer: The correction has been made.

16) P. 10 L. 10 change "amongst other" to" in comparison to

Answer: The correction has been made.

17) P. 10 L. 17 or 0.07 – 0.14 or 0.014-0.07 but not as written

Answer: The correction has been made.

18) P. 14 L. 13 "downstream" or downgradient

Answer: " downgradient " - The correction has been made.

19) P. 14L. 15 "(2015, 2012b)" there is only 1 reference of Hillebrand et al. in the reference list

Answer: The correction has been made: Hilllebrand et al., 2012b was added in the reference list.

20) P. 14 L.18-21. Consider discarding, out of context and does contribute much.

Answer: The correction has been made.

Please also note the supplement to this comment:
https://www.hydrol-earth-syst-sci-discuss.net/hess-2018-426/hess-2018-426-AC1-supplement.pdf

––––––––––––––––––––––––––––––

The upper Sorek Basin monitoring sites and flow and transport simulation domains map, showing monitoring wells EK 1, EK 4, EK 6, EK 11, EK 15, Sorek Creek, Beit Zait Reservoir, and simulation domains. Legend includes Sewage Discharge Site, Collector Pipe, Gauging Station, Tzuba Meteorological Station, Observation Well, Late winter 2011 WL contour (masl), Sorek Creek, Flow Simulation (large) Domain, Transport Simulation (small) Domain, and Soreq Watershed. Inset map shows Study Area, Aquifer Boundaries, Mediterranean coastline, TEL AVIV-YAFO, and JERUSALEM.

**Fig. 1.** The upper Sorek Basin monitoring sites and flow and transport simulation domains (after Dvory et al., 2018a; aquifer boundaries from Dafny, 2009)

[Figure]

Fig. 2. Model conceptual sketch

Time series data for rainfall, dam runoff, sewage, water level, and micro-contaminants.

**Fig. 3.** Time series data observation and calculation (after Dvory et al., 2018a). (A) Tzuba Meteorological station daily precipitation rate; (B) Dam runoff flow; (C) Sewage surface flow; (D) Measured and simu

**Fig. 4.** (A) Observed and simulated BTCs of CBZ and CAF in EK11; (B) Relative concentration variations of CBZ and CAF in EK11 (CBZ data from Dvory et al., 2018a).

[Figure]

**Fig. 5.** Simulated CAF sensitivity to parameters changes (A) the degradation rate and (B) the distribution coefficient. The insets show the effect of parameters on RMSE.

---

## Author Comment (AC2) · 1 Oct 2018

Answers to Reviewer #2 comments:

Thank you for the feedback. Your recommendations were helpful and insightful. All of the comments have been addressed, and the paper was edited accordingly.

General comments:

Figures 1,2 and 3b are taken from the previous manuscript with no or very little modification. In addition to the overlap as indicated by the editor this should be fixed (Figure 1 is probably the most critical one and should be slightly modified, also to

[Figure]

avoid any copyright issues).

Answer: As suggested by the reviewer, the Fig. 1 was modified. In the caption under figure 2 (Figure 3 in the revised manuscript) we indicate that the figure references Dvory et al., 2018a. Figure 3 (Figure 4 in the revised manuscript) includes CAF concentration unlike the previous paper published in WR.

Content-related comments:

1. (p6): For the sake of completeness please add units to the description of the equations.

Answer: The units were added.

2. (p5): The section "Numerical model" would greatly benefit from a conceptual sketch of the model framework (the Water Research paper provides a conceptual model of the hydrogeological system only). The model is quite complex and as the authors have already limited the model description here. A conceptual sketch would also allow readers to understand more aspects of the model without first having to read another paper. This would also help to make the paper stand out slightly more compared to the WR paper.

Answer: As suggested by the reviewer, we added an additional figure (fig. 2) that we hope that will assist the readers to understand the model framework.

3. (p5, line 28): From the description here it is not entirely clear to me if both the 1D and 3D part of the domain are subject to a multi-continuum coupling. In this sense also the terms high and low permeable region are (from a conceptual point of view) associated with different compartments of the aquifer. High permeable regions in the vadose zone are possibly (enlarged) fractures and to a limited degree former conduit systems depending on the long term evolution of the system, while in the 3D part the high permeable regions are commonly the conduits. This should be clarified, possibly also in conjunction with my previous comment to add a conceptual sketch.

Answer: In the revised manuscript we clarify that multi-continuum coupling was done for both the unsaturated and saturated zones, which is also shown in the conceptual sketch (new Figure 2). In the mathematical model, fractures and conduits belongs to one continuum, while porous matrix to another.

4. (p6): In addition to the conceptual sketch I think a figure showing the discretized model domain including boundary conditions (both for the large and small model) would be adequate to be added to the section "Mathematical model setup".

Answer: We present the boundary conditions in Figure 2 (according to the revised version), however we prefer not to show the finite-differences discretization of the domains because it's technical and overloads the figure. The sizes of numerical grids are mentioned in the text.

5. (p7, line 15-19): To what extent does the vadose zone possibly affect the (bulk) dispersivity? The chosen approach is common for saturated systems but may be affected by the vadose zone which imposes an additional transformation/dispersion of the signal. I understand that this is a very difficult topic and would only ask for a brief comment if this might be the case (or not if the authors can clearly rule this out). In this context the authors mention that CBZ is stored in the vadose zone (on page 14, line 11), hence I would expect an influence.

Answer: We agree with the reviewer. The dispersivity parameters could be different for the vadose zone and groundwater, however, with the quality of data we have (breakthrough curves in one observation well) we can only obtain a lumped parameter for both unsaturated and saturated zones. We added the following note in section 3.2 (page 10, lines 11-16): "…These values, calculated in this study, represent combined vadose zone-groundwater model characteristics. Even though the presence of air phase can influence the physico-chemical processes of contaminant transport and transformation given the quality of dataset available (breakthrough curves in one observation well) we can only obtain lumped parameters for both the unsaturated and saturated zones. The

effect of variable water saturation on pollution dispersion and degradation is accounted for by multiplying these parameters by the water content (equations (1) and (2)). " CBZ is stored in the vadose zone mostly in low permeability sites (matrix) and the rate exchange between matrix and conduits influences the transport. We added a clarification on this in section 3.4 (page 15, lines 13-15): "The tail of the low CBZ concentration during the dry season is a result of low saturation in the vadose zone. This reduces the hydraulic conductivity and the exchange between matrix and conduits, resulting in low CBZ transport rates toward the aquifer"

6. (p12): Are the parameters $\lambda$ and Kd defined for both the 3D section and the 1D vadose zone? This information should be added. Please also see my comment 3. Is the degradation of caffeine affected by the presence of an airphase? If this is the case then this should be briefly discussed either here or in the discussion section.

Answer: $\lambda$ and Kd were defined for both vadose zone and the aquifer. We added a remark to this effect in section 2.4.3 (page 7, line 12) and briefly discussed this in section 3.2 (page10, lines11-15)

7. (p13, Fig4): I am a bit confused by both sensitivity analyses but may have missed some information in the manuscript. I would expect an opposite behavior for $\lambda$ as well as Kd. I would expect lower peaks (and low tailing, i.e. generally a decline in mass) for higher values of degradation. The same applies for the distribution coefficient (which to my knowledge is commonly defined as activity of solid/aqueous phase). Here I would expect lower peaks for higher values of Kd as CAF tends to be in an sorped state. In Figure 4a it is difficult to see where the peak of parameter combination 6 is (only the tailing is clearly visible). A different color (gray or colored) for the fitted values (both in A/B) could help to enhance visual clarity.

Answer: Thank you for this important remark. The legend in this figure was wrong. We corrected the legend and the text accordingly. We also changed this figure (Figure 5 in the revised manuscript) to make it more coherent by distinguishing between the

different graphs lines.

8. (p14, line 5): Is this correct? I would expect low background concentrations to be beneficial for the detection of a new signal.

Answer: Yes, it is correct. When the new event has low concentration levels that can occur as a result of surface or subsurface dilution it is more difficult to detect it from previous background concentrations.

Typographic corrections:

I am not a native speaker and can only partially comment on proper grammar. The following are mostly typographic corrections and recommendations to enhance the comprehensibility.

1. (p3, line 6): Maybe rephrase. Do the authors mean that Sorek creek watersheed accounts for 88km2 within the study area or that the Sorek creek study area is 88km2 in size?

Answer: The correction has been made as follows: "The Sorek creek watershed drains approximately 88 km2 in the study area and is located west of the city of Jerusalem, Israel (Fig. 1)"

2. (p3, line 19, 20): This may be journal-specific but commonly only numbers exceeding 12 are spelled out.

Answer: The correction has been made.

3. (p6-7, line 28/1): I assume the authors mean main memory not the CPU cache. Possibly rephrase as "owing to a lack of main memory".

Answer: This part of the sentence was removed from the paper.

4. (p7, line 4): Is the grid becoming finer towards the top or the bottom of the domain? Possibly rephrase to clarify.

Answer: In the revised manuscript we indicate that the grid become finer towards the top of the matrix (the ground surface).

5. (p12, line 11): "...assigning $\lambda = 0$ in the matrix,..."

Answer: The sentence is correct. Thus, we tested a scenario in which CAF degradation is neglected in the matrix.

6. (p13, Fig. 4): Please match the font/fontsize of the insets in A and B (lambda and Kd values)

Answer: The correction has been made.

7. (p14, line 23): "A quasi 3D dual permeability..."

Answer: The correction has been made.

8. (p14, line 25): I think it should be "calibration with monitoring data..."

Answer: The correction has been made.

Please also note the supplement to this comment:
https://www.hydrol-earth-syst-sci-discuss.net/hess-2018-426/hess-2018-426-AC2-supplement.pdf
* * *
**Fig. 1.** The upper Sorek Basin monitoring sites and flow and transport simulation domains
(after Dvory et al., 2018a; aquifer boundaries from Dafny, 2009)

[Figure]

**Flow simulation**

Hydraulic parameters & flow BC for small domain

**LARGE DOMAIN**
(1600×1400 m)
with a COARSE grid

**Flow and CBZ transport simulation**

**SMALL DOMAIN**
(590×460 m)
with a FINE grid

**Flow and CAF transport simulation**

Sorek creek

Vadose zone
1D flow and transport

**Dynamic phreatic surface**
Coupling 1D and 3D equations

Groundwater
3D flow and transport

Dual permeability model — Matrix-slow flow

Conduit-fast flow

Exchange by water and solute

**Boundary conditions (BC)**
1 Zero water and solute fluxes
2 Variable head
3 Variable water flux
4 Variable water flux (infiltration along the creek and rain-evaporation elsewhere), and variable solute concentration along the creek

**Fig. 2.** Model conceptual sketch

[Figure]

**Fig. 3.** Time series data observation and calculation (after Dvory et al., 2018a). (A) Tzuba Meteorological station daily precipitation rate; (B) Dam runoff flow; (C) Sewage surface flow; (D) Measured and simu

[Figure]

**Fig. 4.** (A) Observed and simulated BTCs of CBZ and CAF in EK11; (B) Relative concentration variations of CBZ and CAF in EK11 (CBZ data from Dvory et al., 2018a).

**Fig. 5.** Simulated CAF sensitivity to parameters changes (A) the degradation rate and (B) the distribution coefficient. The insets show the effect of parameters on RMSE.

**Supplement:**

**Answers to Reviewer #2 comments:**

Thank you for the feedback. Your recommendations were helpful and insightful. All of the comments have been addressed, and the paper was edited accordingly.

**General comments**

**Figures 1,2 and 3b are taken from the previous manuscript with no or very little modification. In addition to the overlap as indicated by the editor this should be fixed (Figure 1 is probably the most critical one and should be slightly modified, also to avoid any copyright issues).**

**Answer:** As suggested by the reviewer, the Fig. 1 was modified. In the caption under figure 2 (Figure 3 in the revised manuscript) we indicate that the figure references Dvory et al., 2018a. Figure 3 (Figure 4 in the revised manuscript) includes CAF concentration unlike the previous paper published in WR.

**Content-related comments:**

**1. (p6): For the sake of completeness please add units to the description of the equations.**

**Answer:** The units were added.

**2. (p5): The section "Numerical model" would greatly benefit from a conceptual sketch of the model framework (the Water Research paper provides a conceptual model of the hydrogeological system only). The model is quite complex and as the authors have already limited the model description here. A conceptual sketch would also allow readers to understand more aspects of the model without first having to read another paper. This would also help to make the paper stand out slightly more compared to the WR paper.**

**Answer:** As suggested by the reviewer, we added an additional figure (fig. 2) that we hope that will assist the readers to understand the model framework.

**3. (p5, line 28): From the description here it is not entirely clear to me if both the 1D and 3D part of the domain are subject to a multi-continuum coupling. In this sense also the terms high and low permeable region are (from a conceptual point of view) associated with different compartments of the aquifer. High permeable regions in the vadose zone are possibly (enlarged) fractures and to a limited degree former conduit systems depending on the long term evolution of the system, while in the 3D part the high permeable regions are commonly the conduits. This should be clarified, possibly also in conjunction with my previous comment to add a conceptual sketch.**

**Answer:** In the revised manuscript we clarify that multi-continuum coupling was done for both the unsaturated and saturated zones, which is also shown in the conceptual sketch (new Figure 2). In the mathematical model, fractures and conduits belongs to one continuum, while porous matrix to another.

**4. (p6): In addition to the conceptual sketch I think a figure showing the discretized model domain including boundary conditions (both for the large and small model) would be adequate to be added to the section "Mathematical model setup".**

**Answer:** We present the boundary conditions in Figure 2 (according to the revised version), however we prefer not to show the finite-differences discretization of the domains because it's technical and overloads the figure. The sizes of numerical grids are mentioned in the text.

**5. (p7, line 15-19): To what extent does the vadose zone possibly affect the (bulk) dispersivity? The chosen approach is common for saturated systems but may be affected by the vadose zone which imposes an additional transformation/dispersion of the signal. I understand that this is a very difficult topic and would only ask for a brief comment if this might be the case (or not if the authors can clearly rule this out). In this context the authors mention that CBZ is stored in the vadose zone (on page 14, line 11), hence I would expect an influence.**

**Answer:** We agree with the reviewer. The dispersivity parameters could be different for the vadose zone and groundwater, however, with the quality of data we have (breakthrough curves in one observation well) we can only obtain a lumped parameter for both unsaturated and saturated zones. We added the following note in section 3.2 (page 10, lines 11-16): "…These values, calculated in this study, represent combined vadose zone-groundwater model characteristics. Even though the presence of air phase can influence the physico-chemical processes of contaminant transport and transformation given the quality of dataset available (breakthrough curves in one observation well) we can only obtain lumped parameters for both the unsaturated and saturated zones. The effect of variable water saturation on pollution dispersion and degradation is accounted for by multiplying these parameters by the water content (equations (1) and (2)). "

CBZ is stored in the vadose zone mostly in low permeability sites (matrix) and the rate exchange between matrix and conduits influences the transport. We added a clarification on this in section 3.4 (page 15, lines 13-15):

"The tail of the low CBZ concentration during the dry season is a result of low saturation in the vadose zone. This reduces the hydraulic conductivity and the exchange between matrix and conduits, resulting in low CBZ transport rates toward the aquifer"

**6. (p12): Are the parameters λ and Kd defined for both the 3D section and the 1D vadose zone? This information should be added. Please also see my comment 3. Is the degradation of caffeine affected by the presence of an airphase? If this is the case then this should be briefly discussed either here or in the discussion section.**

**Answer:** λ and Kd were defined for both vadose zone and the aquifer. We added a remark to this effect in section 2.4.3 (page 7, line 12) and briefly discussed this in section 3.2 (page10, lines11-15)

**7. (p13, Fig4): I am a bit confused by both sensitivity analyses but may have missed some information in the manuscript. I would expect an opposite behavior for λ as well as Kd. I would expect lower peaks (and low tailing, i.e. generally a decline in mass) for higher values of degradation. The same applies for the distribution coefficient (which to my knowledge is commonly defined as activity of solid/aqueous phase). Here I would expect lower peaks for higher values of Kd as CAF tends to be in an sorped state. In Figure 4a it is difficult to see where the peak of parameter combination 6 is (only the tailing is clearly visible). A different color (gray or colored) for the fitted values (both in A/B) could help to enhance visual clarity.**

**Answer:** Thank you for this important remark. The legend in this figure was wrong. We corrected the legend and the text accordingly. We also changed this figure (Figure 5 in the revised manuscript) to make it more coherent by distinguishing between the different graphs lines.

**8. (p14, line 5): Is this correct? I would expect low background concentrations to be beneficial for the detection of a new signal.**

**Answer:** Yes, it is correct. When the new event has low concentration levels that can occur as a result of surface or subsurface dilution it is more difficult to detect it from previous background concentrations.

**Typographic corrections:**

**I am not a native speaker and can only partially comment on proper grammar. The following are mostly typographic corrections and recommendations to enhance the comprehensibility.**

**1. (p3, line 6): Maybe rephrase. Do the authors mean that Sorek creek watersheed accounts for 88km2 within the study area or that the Sorek creek study area is 88km2 in size?**

**Answer:** The correction has been made as follows: "The Sorek creek watershed drains approximately 88 km2 in the study area and is located west of the city of Jerusalem, Israel (Fig. 1)"

**2. (p3, line 19, 20): This may be journal-specific but commonly only numbers exceeding 12 are spelled out.**

**Answer:** The correction has been made.

**3. (p6-7, line 28/1): I assume the authors mean main memory not the CPU cache. Possibly rephrase as "owing to a lack of main memory".**

**Answer:** This part of the sentence was removed from the paper.

**4. (p7, line 4): Is the grid becoming finer towards the top or the bottom of the domain? Possibly rephrase to clarify.**

**Answer:** In the revised manuscript we indicate that the grid become finer towards the top of the matrix (the ground surface).

**5. (p12, line 11): "...assigning λ = 0 in the matrix,..."**

**Answer:** The sentence is correct. Thus, we tested a scenario in which CAF degradation is neglected in the matrix.

**6. (p13, Fig. 4): Please match the font/fontsize of the insets in A and B (lambda and Kd values)**

.**Answer:** The correction has been made

**7. (p14, line 23): "A quasi 3D dual permeability..."**

**Answer:** The correction has been made.

**8. (p14, line 25): I think it should be "calibration with monitoring data..."**

**Answer:** The correction has been made.

[revised manuscript text omitted]

---

## Referee Report (RR1)

**Review (revision 2): Caffeine vs Carbamazepine as indicators for wastewater pollution in a karst aquifer**

**November 10, 2018**

The authors addressed all of my remarks and I recommend publication. The manuscript is short and concise and reads very well. Please see below for some minor remarks/corrections. In case this is not part of the technical editing of the journal, I would recommend to replace blanks between values and units by thin spaces, e.g. 10 m instead of 10 m.

**Typographic corrections:**

1. (Fig1): either remove the term "Legend" or slightly move it to the right.

2. (p3, line 8): The sentence seems to miss a word: "... from the dam downstream may occur"?

3. (p5, line 5): °C

4. (p6, line 14): "... while a zero concentration gradient..."

5. (p6, line 19): "... incorporate a/the 1D Richards' equation..."

6. (p8, line 7): $t_j$ ? ... j as subscript.

7. (p8, line 12): "... to the calibrated model RMSE"?

8. (p10, line 28): 0.014-0.07 $d^{-1}$ ?

9. (p15, line 9): "... performed ..."

---

## Author Response (AR2)

We wish to thank both reviewers for the feedback regarding typographic corrections. All of the comments have been addressed, and the paper was edited accordingly.

**Answers to Reviewer #1 Report #2 comments:**

**One technical-typo in line 27 at page 6, the end of the sentence "owing …memory" was probably deleted in track changes but still appears as deleted text in the published revision, course. No need for another iteration with the authors for that.**

**Answer:** The correction has been made.

**Answers to Reviewer #2 Report #2 comments:**

**1. (Fig1): either remove the term "Legend" or slightly move it to the right.**

**Answer:** The correction has been made.

**2. (p3, line 8): The sentence seems to miss a word: "… from the dam downstream may occur"?**

**Answer:** The correction has been made.

**3. (p5, line 5): $^\circ$C**

**Answer:** The correction has been made.

**4. (p6, line 14): "… while a zero concentration gradient…"**

**Answer:** The correction has been made.

**5. (p6, line 19): "… incorporate a/the 1D Richards' equation…"**

**Answer:** The correction has been made.

**6. (p8, line 7): $t_j$ ? … j as subscript.**

**Answer:** The correction has been made.

**7. (p8, line 12): "… to the calibrated model RMSE"?**

**Answer:** The correction has been made.

**8. (p10, line 28): 0.014-0.07 $d^{-1}$ ?**

**Answer:** The correction has been made.

**9. (p15, line 9): "… performed …"**

**Answer:** The correction has been made.